# Neuronal activity in dorsal anterior cingulate cortex during economic choices under variable action costs

Xinying Cai[1,2,3,4]*, Camillo Padoa-Schioppa[1,5,6]

[1]Department of Neuroscience, Washington University in St Louis, Saint Louis, United States; [2]NYU Shanghai, Shanghai, China; [3]Shanghai Key Laboratory of Brain Functional Genomics (Ministry of Education), School of Psychology and Cognitive Science, East China Normal University, Shanghai, China; [4]NYU-ECNU Institute of Brain and Cognitive Science at NYU Shanghai, Shanghai, China; [5]Departments of Economics, Washington University in St Louis, St Louis, China; [6]Biomedical Engineering, Washington University in St Louis, St Louis, China

**ABSTRACT** The role of the dorsal anterior cingulate cortex (ACCd) in decision making has often been discussed but remains somewhat unclear. On the one hand, numerous studies implicated this area in decisions driven by effort or action cost. On the other hand, work on economic choices between goods (under fixed action costs) found that neurons in ACCd encoded only post-decision variables. To advance our understanding of the role played by this area in decision making, we trained monkeys to choose between different goods (juice types) offered in variable amounts and with different action costs. Importantly, the task design dissociated computation of the action cost from planning of any particular action. Neurons in ACCd encoded the chosen value and the binary choice outcome in several reference frames (chosen juice, chosen cost, chosen action). Thus, this area provided a rich representation of post-decision variables. In contrast to the OFC, neurons in ACCd did not represent pre-decision variables such as individual offer values in any reference frame. Hence, ongoing decisions are unlikely guided by ACCd. Conversely, neuronal activity in this area might inform subsequent actions.

*For correspondence:
xinying.cai@nyu.edu

Competing interest: The authors declare that no competing interests exist.

## Introduction

Economic choices involve the computation and comparison of subjective values, which integrate multiple attributes relevant to the decision. In particular, obtaining rewards often entails different levels of physical effort. We previously showed that economic choices under variable action costs can take place in a non-spatial representation (goods space) (*Cai and Padoa-Schioppa, 2019*). In the experiments, monkeys chose between two juices offered in variable amounts and associated with different action costs. The task design dissociated the computation of action costs from action planning. A group of neurons in the orbitofrontal cortex (OFC) encoded offer values integrating juice type, juice amount, and action cost. Moreover, another group of cells in OFC encoded the choice outcome before presentation of the action targets, indicating that decisions were made in goods space. However, it remained unclear whether and how other brain regions contribute to this choice process. When goods available for choice bear different action costs, the decision necessarily reflects aspects of the action. Thus, signals from the ACCd and/or other motor regions could provide an input for the computation of subjective values in OFC.

There are several reasons to believe that ACCd might be important for choices under variable action cost. First, both anatomic and functional evidence indicates that ACCd plays a role in linking

actions and rewards (*Bush et al., 2000*; *Calderazzo et al., 2021*; *Hadland et al., 2003*; *Hayden and Platt, 2010*; *Paus, 2001*). Second, ACCd has been implicated in various types of decision making, including strategic decision making, social decision making, and cost-benefit tradeoff (*Klein-Flügge et al., 2016*; *Lee et al., 2007*; *Rangel and Hare, 2010*; *Rushworth et al., 2007*; *Wallis and Kennerley, 2010*). More importantly, previous studies found that neuronal activity in this region integrates expected reward and action cost (*Kennerley et al., 2009*; *Kennerley and Wallis, 2009*) and reflects effort-discounted subjective values in an action-based reference frame (*Hosokawa et al., 2013*). In rodent studies, lesion of the medial frontal cortex (MFC, roughly homologous to the primate ACCd) disrupts reward and effort tradeoff (*Rudebeck et al., 2006*; *Walton et al., 2002*). Furthermore, a recent study showed that chemogenetic modulation of the rodent ACCd altered the animal's effort-exerting behavior in conditions that require a choice, but not when choices were not required (*Hart et al., 2020*).

In a previous study, we examined the neuronal representations in ACCd in an economic choice task with equal action cost (*Cai and Padoa-Schioppa, 2012*). We discovered that ACCd only encodes post-decision variables, namely, *chosen value*, *chosen juice,* and *movement direction*. However, based on its engagement in effort-based decisions, ACCd may contribute to the computation of *offer values* when decisions require the integration of action costs. To test this possibility, in the current study, we introduced action cost as an additional value attribute. The action cost was operationalized by requiring the animal to make saccades of different amplitudes. Critically, the action cost was indicated to the animal before and independent of the instruction associated with any particular action. We report the following results. First, action costs taxing the oculomotor system induced appreciable behavioral effects as the animals 'choices' were biased in favor of low-cost offers. Second, contrary to reasonable predictions, neurons in ACCd did not encode *offer values* in any reference frame. Conversely, neurons in ACCd encoded different variables associated with the choice outcome. Third, in contrast to OFC, where neurons encoded choice outcome in juice-based and cost-based reference frames with approximately equal strength, the encoding of choice outcomes in ACCd was heavily skewed in favor of a cost-based reference frame. Taken together, these results extend previous observations on good-based choices and support the hypothesis that ACCd plays a role in guiding motor systems based on reward and effort associated with the choice outcomes.

## Results
### Experimental design and choice patterns

We designed a task in which animals chose between two juices offered in variable amounts (*Figure 1A*). We dissociated the spatial location of the offers from the saccades necessary to obtain them and we introduced a delay between the presentation of the offers and the saccade targets. Moreover, offers were associated with radial eye movements in different directions, and different saccade amplitudes imposed variable action costs. We reasoned that if the initial fixation point is straight ahead of the subject, the action cost associated with an eye movement is essentially independent of the saccade direction (isotropic) and only depends on the saccade amplitude. Each offer provided information about all the attributes of value – juice type indicated by symbol color, quantity indicated by symbol number, and action cost indicated by symbol shape, while the animal was unable to plan the saccade necessary to obtain the offer. For any given trial, one 'offer' was defined by a juice type, its quantity, and its action cost. An 'offer type' was defined by two offers (e.g., [1A+:3B−]), in which '−' indicates high action cost (long saccade) and '+' indicates low action cost (short saccade). A 'trial type' was defined by two offers and a choice (e.g., [1A+:3B−, A]). Thus, a 'trial type' encompasses task-related factors in goods space.

Behavioral evidence indicated that the experimental manipulation was effective in producing an action cost. *Figure 1B* illustrates the choice pattern recorded in one representative session. Trials were divided into two groups depending on whether juice A was offered at low cost or at high cost. The gray sigmoid is displaced to the right, indicating that the relative value of juice A was higher when juice A was offered at low cost. This effect was consistent across sessions for both monkeys. Our data set included 141 behavioral sessions from both animals. For a quantitative analysis of choice patterns, we constructed a logistic model that provided measures for the relative value of the two juices ($\rho$), the difference in action cost ($\xi$), the choice hysteresis related to the chosen juice ($\eta$) and to the

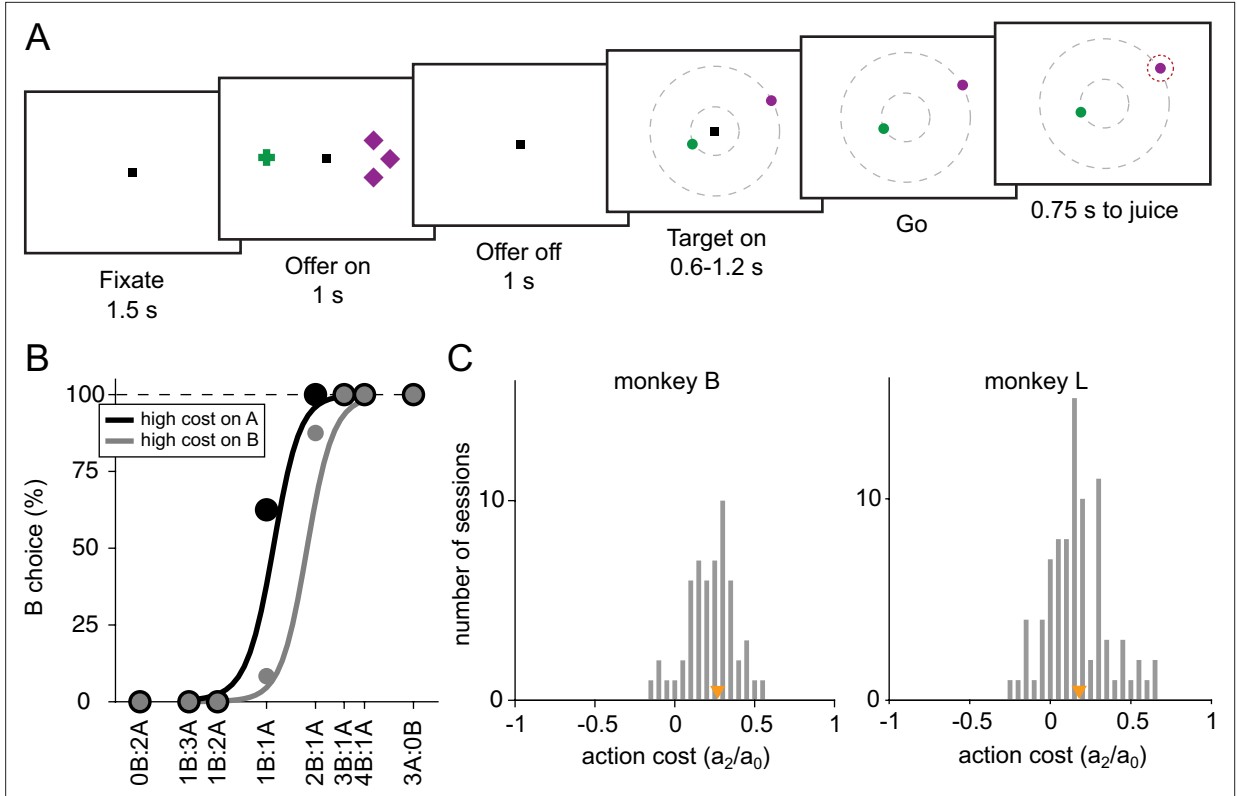

**Figure 1.** Experimental design and behavioral analysis. (**A**) At the beginning of the trial, the monkey fixated a center point on the monitor. After 1.5 s, two offers appeared to the left and right of the fixation point. The offers were represented by sets of color symbols, with the color indicating the juice type, the number of symbols indicating juice amount and the shape of the symbols indicating the action cost associated with the offer (crosses, low cost; diamonds high cost). The offers remained on the monitor for 1 s, then they disappeared. The monkey continued fixating the center point for another 1 s. At the end of this delay, two saccade targets (two color dots) appeared. The two saccade targets were located on two (invisible) concentric rings centered on the fixation point. The animal maintained fixation for a randomly variable delay (0.6–1.2 s) before the center fixation point was extinguished ('go' signal), at which point the monkey indicated its choice with a saccade. (**B**) Choice patterns, one session. The percentage of B choices is plotted against the ratio $q_B$:$q_A$, where $q_A$ and $q_B$ are quantities of juice A and juice B, respectively. Trials were separated in two groups depending on the level of action cost for juice A. The choice pattern obtained when juice A had a high cost (black) was displaced to the left (lower indifference point) compared to the choice pattern obtained when juice A had a low cost (gray). The regression lines were obtained with a simplified version of Equation (1) in which terms $a_3$ to $a_6$ were removed. The action cost can be measured as $\xi = a_2/a_0$ (Equation 1). (**C**) Histogram of action costs across 56 sessions for monkey B (median($\xi$) = 0.27, p < 10$^{-9}$, Wilcoxon signed-rank test) and 85 sessions for monkey L (median($\xi$) = 0.18, p < 10$^{-11}$, Wilcoxon signed-rank test). Orange triangles indicate median values.

The online version of this article includes the following source data for figure 1:

**Source data 1.** Source data for **Figure 1c**.

chosen cost ($\phi$), and the spatial biases related to the offer position ($\delta$) and to the target position ($\varepsilon$) (see Materials and methods, Equation 1). We first examined the distribution of $\xi$ across sessions. The difference in saccade amplitude had a significant effect on choices in both animals (**Figure 1C**, monkey B, median($\xi$) = 0.27, p < 10$^{-9}$; monkey L, median($\xi$) = 0.18, p < 10$^{-11}$; Wilcoxon signed-rank test).

In addition to action cost, we also examined other factors that may influence the animal's choice. These factors include the monkey's tendency to choose on any given trial the same juice chosen and received in the previous trial (choice hysteresis; **Padoa-Schioppa, 2013**). This effect was quantified by the normalized coefficient $\eta$ (Equation 1, median($\eta$) = 0.16, p < 10$^{-19}$; Wilcoxon signed-rank test). We also tested whether the animal's choices were affected by the cost associated with the offer chosen in previous trial (cost hysteresis). This effect, quantified by the normalized coefficient $\phi$ (Equation 1), was not significant across sessions (median($\phi$) = –0.008, p = 0.58; Wilcoxon signed-rank test). The normalized coefficient δ quantified offer-based spatial biases. Across sessions, this effect was statistically significant but rather small (median(δ) = –0.03, p = 0.04; Wilcoxon signed-rank

test). Finally, the normalized coefficient ε quantified target-based spatial biases. Similar to offer-based spatial biases, this effect was statistically significant but very small in amplitude (median(δ) = 0.017, p = 0.027; Wilcoxon signed-rank test).

## Encoding of choice outcome in multiple reference frames

We recorded and analyzed the activity of 688 neurons from the ACCd of two monkeys (B, 318 cells; L, 370 cells). Firing rates were analyzed in nine time windows aligned with different behavioral events (see Materials and methods). A 'neuronal response' was defined as the activity of one neuron in one time window as a function of the trial type. Inspection of individual responses (see Materials and methods) indicated that many neurons in ACCd were modulated by the trial type and encoded variables related to the choice outcome. More specifically, ACCd neurons appeared to encode decision variables in the frames of reference defined by the juice type and/or the action cost. *Figure 2* illustrates a few examples. The response in *Figure 2A*, recorded in the late-delay time window, varied as a linear function of the value of chosen offer (variable *chosen value*). Conversely, the response in *Figure 2B*, recorded in the post-juice time window, was low when the animal chose juice A and high when the animal chose juice B (variable *chosen juice*). Meanwhile, the response in *Figure 2C*, recorded in the pre-go time window, was modulated by both chosen cost (low when the animal chooses the low-cost offer and high when the animal chooses the high-cost offer) and the location of chosen target (low when the chosen target is in the ipsilateral hemifield and high when the chosen target is in the contralateral hemifield).

Further inspection revealed that many neurons in the ACCd also encode the spatial location of the chosen offer or chosen target. For example, the response in *Figure 3A* (late-delay time window) was roughly binary – low when the chosen offer was located on the ipsilateral hemifield and high the chosen offer was on the contralateral hemifield. It did not vary significantly with the *chosen value*. Similarly, the response in *Figure 3B* (pre-go time window) was low when the chosen target was in the ipsilateral hemifield and high when the chosen target was in the contralateral hemifield, and did not vary with the *chosen value*.

For a population level statistical analysis, we proceeded in steps. First, we submitted each neuronal response to two 3-way analysis of variances (ANOVAs) (factors [trial type × offer A location × target A location]; factors [trial type × chosen offer location × chosen target location]; see Materials and methods). We imposed a significance threshold p < 0.001. Responses that passed this criterion for at least one factor in one of the two ANOVAs were identified as 'task related' and included in subsequent analyses. As detailed in *Table 1*, out of the total 688 neurons, many neurons were modulated by the trial type (172 cells = 25%) and/or the chosen target location (108 cells = 15.7%), while few cells were modulated by the spatial configuration of the offers (5 cells = 0.7%), the spatial configuration of the targets (25 cells = 3.6%), or the location of the chosen offer (38 cells = 5.5%). Overall, 249 cells (36.2%) were modulated by at least one factor. Subsequent analyses were restricted to this population.

In a previous study that did not dissociate the spatial location of the offer from that of the saccade target, we found that neuronal responses in ACCd often reflected the chosen side (*Cai and Padoa-Schioppa, 2012*). Here, we examined neuronal responses with a series of analysis of covariances (ANCOVAs), continuous variables were those defined in *Table 2*. As a covariate, we used the side of the chosen offer for early time windows (from post-offer to pre-target time windows), and the side of the chosen target for late time windows (from post-target to post-juice time windows; see Materials and methods).

*Figure 4* illustrates the results obtained for the population. *Figure 4A* indicates the number of responses explained by each variable in each time window. Notably, each response could be explained by more than one variable and thus could contribute to multiple bins in this panel. *Figure 4B* illustrates a complementary account. Here, each response was assigned to the variable that provided the best fit. In early time windows, the dominant variables were *chosen value* and *chosen location only*. In late time windows, after target presentation and upon juice delivery, the dominant variables were *chosen value*, *chosen target location only*, *chosen cost,* and *chosen juice*. Two procedures – stepwise and best-subset – were used to identify the variables that best explained the neuronal data set (see Materials and methods). Variables were selected across all time windows. Both procedures selected variables *chosen value*, *chosen cost*, *chosen juice*, *chosen offer location only,* and *chosen*

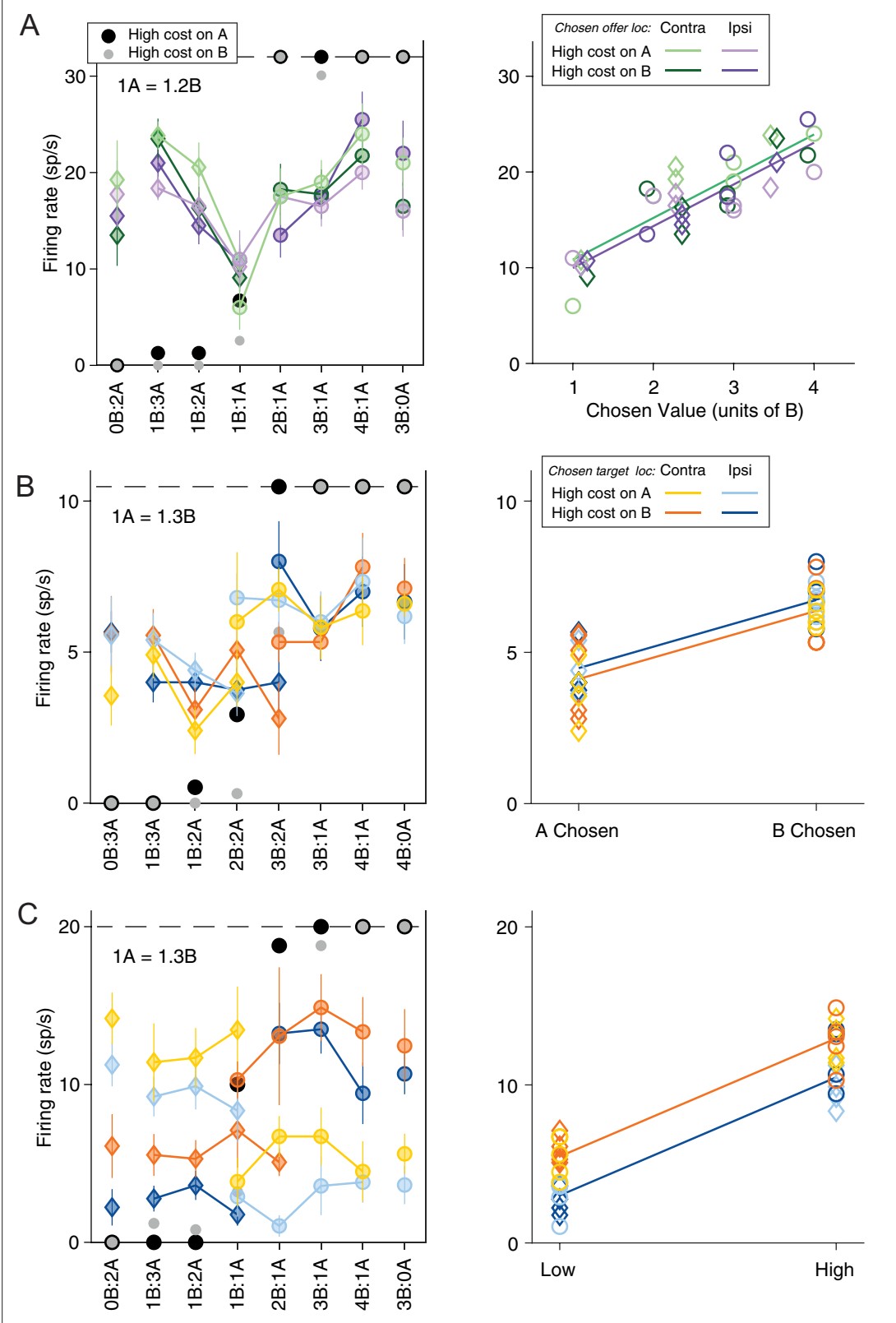

**Figure 2.** Neuronal encoding of variables in commodity and cost-based reference frames. (**A**) Response encoding the chosen value (late-delay time window). In the left panel, the x-axis represents different offer types ranked by ratio $q_B:q_A$. Black and gray symbols represent the percentage of 'B' choices for A− and A+ trials, respectively. Color symbols represent the neuronal firing rate, with diamonds and circles indicating trials in which the monkey chose juice A and juice B, respectively. Green and purple indicate the chosen offer located on the contralateral or ipsilateral hemifield and

*Figure 2 continued on next page*

*Figure 2 continued*

light and dark shade indicate A+ and A– trials, respectively. Error bars indicate SEM. In the right panel, the same neuronal response is plotted against the variable chosen value and chosen offer location. The parallel green and purple lines are derived from an analysis of covariance (ANCOVA) using the location of the chosen offer as a covariate (parallel lines). This response encoded the variable *chosen value* ($R^2 = 0.75$, $p_{chosen\ value} < 10^{-10}$) but was not modulated by the chosen offer location ($p_{chosen\ offer\ loc} = 0.30$). (**B**) Response encoding the chosen juice (post-juice time window). All conventions are as in (**A**) except that the blue and orange color indicate chosen target location. In the right panel, the response is plotted against the variable chosen juice and chosen target location. The parallel blue and orange lines are derived from ANCOVA ($R^2 = 0.64$, $p_{chosen\ value} < 10^{-7}$, $p_{chosen\ target\ loc} = 0.24$). (**C**) Response conjunctively encoding the chosen cost and chosen target location (pre-go time window). All conventions are as in (**C**). In the right panel, the response is plotted against the variable chosen cost and chosen target location. The parallel blue and orange lines are derived from an ANCOVA ($R^2 = 0.90$, $p_{chosen\ cost} < 10^{-16}$, $p_{chosen\ target\ loc} < 10^{-5}$).

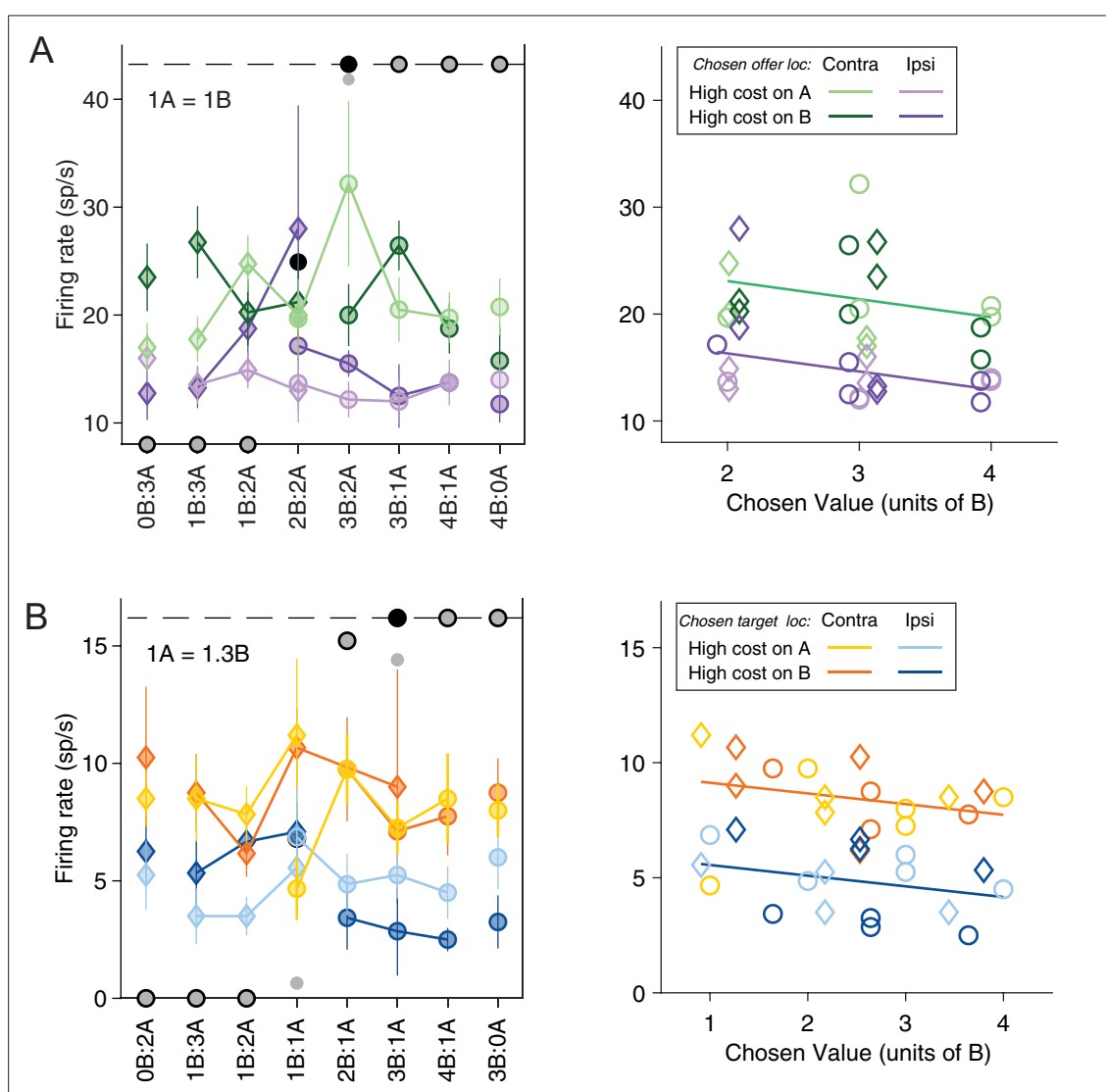

**Figure 3.** Neuronal encoding of variables in spatial and action-based reference frames. (**A**) Response encoding the variable *chosen offer location only* (late-delay time window). All conventions are as in *Figure 2A*. In the right panel, the neuronal response is plotted against the *chosen value* and *chosen offer location*. The parallel green and purple lines are derived from an analysis of covariance (ANCOVA) ($R^2 = 0.49$, $p_{chosen\ value} = 0.070$, $p_{chosen\ offer\ loc} < 10^{-5}$). (**B**) Response encoding the variable *chosen target location only* (pre-go time window). All conventions are as in (**A**) except that the blue and orange color indicate chosen target location. In the right panel, the response is plotted against chosen value and chosen target location. The parallel blue and orange lines are derived from an ANCOVA ($R^2 = 0.62$, $p_{chosen\ value} = 0.10$, $p_{chosen\ target\ loc} < 10^{-7}$).

**Table 1.** A total of 688 cells were recorded from dorsal anterior cingulate cortex (ACCd) and included in this analysis.

The table reports the results of two 3-way analysis of variances (ANOVAs) (factors [trial type × offer A location × target A location]; factors [trial type × chosen offer location × chosen target location]. Each column represents one factor, each row represents one time window, and numbers represent the number of cells significantly modulated by the corresponding factor ($p < 0.001$). Factor trial type is common to the two ANOVAs. The bottom row indicates, for each factor, the number of cells that passed the criterion in at least one of the nine time windows.

| | Trial type | Offer A location | Target A location | Chosen offer location | Chosen target location |
|---|---|---|---|---|---|
| Pre-offer | 1 | 0 | 0 | 1 | 1 |
| Post-offer | 37 | 1 | 1 | 6 | 1 |
| Late-delay | 55 | 0 | 1 | 17 | 0 |
| Mem-delay | 38 | 1 | 0 | 8 | 1 |
| Pre-target | 34 | 0 | 0 | 7 | 1 |
| Post-target | 42 | 0 | 10 | 0 | 28 |
| Pre-go | 35 | 1 | 6 | 1 | 37 |
| Pre-juice | 39 | 0 | 8 | 2 | 61 |
| Post-juice | 29 | 2 | 1 | 1 | 21 |
| At least 1 | 172 | 5 | 25 | 38 | 108 |

The online version of this article includes the following source data for table 1:

**Source data 1.** Source data for *Table 1*.

*target location only*. **Figure 5** illustrates the percentage of neurons encoding each of the selected variables across different time windows.

To summarize, confirming previous results, neurons in ACCd encoded task-relevant variables associated with only the choice outcome. Remarkably, choice outcomes were encoded in all relevant reference frames – juice-, cost-, spatial-, and action-based. In contrast, we did not find any offer value signal in any of these reference frames. Moreover, few neurons in ACCd encode variables *value ratio* and *cost/benefit conflict*, which essentially capture the decision difficulty. Overall, this result demonstrates a prevailing coding of post-decision variables in ACCd.

## Dimensional integration in chosen value cells

Previous studies showed that the activity of *chosen value* cells in the OFC depended on the juice type, juice quantity, and action costs. Here, we assessed whether *chosen value* cells in the ACCd also reflected the action cost. To do so, we defined two variants of the variable *chosen value* (juice) – one cost-affected and one cost-independent. We assessed which variant better fit neuronal responses. For each response, we considered the two $R^2$ and we computed the difference $\Delta R^2 = R^2_{\text{cost-affected}} - R^2_{\text{cost-independent}}$ and we examined the distribution for $\Delta R^2$ across the population. We did not want to bias the results in favor of either variant. Thus, for this analysis we identified neuronal responses encoding the *chosen value* (juice) as follows. For each response, we considered the two $R^2$ obtained from the two variants of *chosen value* variable both 'explaining' the response ($p < 0.05$), and we assigned the maximum $R^2$ to the response. We then assigned each response to one of the selected variables accordingly. As any neuron could be tuned in multiple time windows, we aggregated the *chosen value (juice)* coding responses in the time windows after offer presentation but before target presentation (including post-offer, late-delay, mem-delay, and pre-target windows) and those in the time windows after target presentation (including post-target, pre-go, pre-juice, and post-juice windows) and analyzed them separately.

We discovered that in the post-offer time windows, the distribution of $\Delta R^2$ was significantly biased toward negative values (mean($\Delta R^2$) = –0.013, $p < 10^{-8}$; **Figure 6**). Thus, *chosen value* responses integrate only juice type and quantity in the early time windows before action planning is possible. In

**Table 2.** Defined variables.

In any given trial, $q_A$ and $q_B$ were, respectively, the quantities of juice A and juice B offered to the animal, $\rho$ was the relative value of the two juices, and $\xi$ was the action cost. Parameters $\rho$ and $\xi$ were obtained from the logistic regression (Equation 1). The variable spatial congruence was set = 1 ( = 0) if the offer and the saccade target associated with a given juice (same color) were presented in the same (opposite) hemifield. Variables *chosen offer loc only* and *chosen target loc only* provided the best explanation (highest $R^2$) only if it provided the sole explanation (see Materials and methods).

| | Collapsed variable | Variable | Definition | Reference frame |
|---|---|---|---|---|
| 1 | Offer value (juice) | Offer value A | $\rho\ q_A + \xi\ \delta_{juice\ A,+}$ | Juice |
| 2 | | Offer value B | $q_B + \xi\ \delta_{juice\ B,+}$ | Juice |
| 3 | | Chosen juice | 1 if juice B is chosen, 0 if juice A is chosen | Juice |
| 4 | | Offer value − | Offer value A if A is high cost, offer value B if B is high cost | Cost |
| 5 | Offer value (cost) | Offer value + | Offer value A if A is low cost, offer value B if B is low cost | Cost |
| 6 | | Chosen cost | 1 if low-cost offer is chosen, 0 if high-cost offer is chosen | Cost |
| 7 | Offer value (location) | Offer value L | Value of the juice offered on the left | Visual |
| 8 | | Offer value R | Value of the juice offered on the right | Visual |
| 9 | | Offer value target L | Value of the juice associated with target in the left hemifield | Action |
| 10 | Offer value (target) | Offer value target R | Value of the juice associated with target in the right hemifield | Action |
| 11 | | Cost of A | 1 if offer A is low cost, 0 if offer A is high cost | |
| 12 | | Offer A location | 1 if juice A is offered on left, 0 if juice A is offered on right | |
| 13 | | Target A location | 1 if target A is in left hemifield, 0 if target A is in right hemifield | |
| 14 | | Offer+ location | 1 if low-cost offer is on the left, 0 if low cost offer is on the right | |
| 15 | | Target+ location | 1 if low-cost target is in the left hemifield, 0 otherwise | |
| 16 | | Spatial congruence | 1 if offers and targets are spatially congruent, 0 otherwise | |
| 17 | | Chosen value | Offer value A if juice A chosen, offer value B if juice B chosen | |
| 18 | | Value ratio | Other/chosen value | |
| 19 | | Cost/benefit conflict | Cost of A × sign(offer value A – offer value B). 1 if there is conflict, 0 otherwise | |
| 20 | | Chosen offer loc only | 1 if the chosen offer is on the left, 0 otherwise | Visual |
| 21 | | Chosen target loc only | 1 if the chosen target is in the left hemifield, 0 otherwise | Action |

the post-target time windows, the distribution of $\Delta R^2$ was not statistically different from 0 (mean($\Delta R^2$) = –0.0034, p = 0.25; *Figure 6*). Therefore, we cannot reject the null hypothesis that action cost was integrated by *chosen value* responses later in the trial. Moreover, the difference across early and late time windows was marginally significant (mean ($\Delta R^2$) = 0.0096, p = 0.052; *Figure 6*). Such pattern is consistent with our previous observation in the OFC where *chosen value* responses progressed from integrating only juice type and quantity to integrating all three determinants.

**Figure 4.** Population summary of analysis of covariance (ANCOVA) (all time windows). (**A**) Explained responses. Row and columns represent, respectively, time windows and variables. In each location, the number indicates the number of responses explained by the corresponding variable in that time window. For example, chosen value (juice) explained 34 responses in the post-offer time window. The same numbers are also represented in gray scale. Note that each response could be explained by more than one variable and thus could contribute to multiple bins in this panel. (**B**) Best fit. In

*Figure 4 continued on next page*

*Figure 4 continued*

each location, the number indicates the number of responses for which the corresponding variable provided the best fit (highest R$^2$) in that time window. For example, chosen value (juice) provided the best fit for 41 responses in the late-delay time window. The numerical values are also represented in gray scale. In this plot, each response contributes to at most one bin.

The online version of this article includes the following source data for figure 4:

**Source data 1.** Source data for *Figure 4*.

## Contrasting neuronal encoding in OFC and ACCd

Since our previous study on the OFC used the same choice task and animals used here, we had the opportunity to contrast the neuronal representations in OFC and ACCd, to probe the differential contributions of these two areas to economic choice under variable action costs. First, we noted that neurons in the OFC and ACCd do not encode the same set of variables. In particular, pre-decision variables such as *offer value (juice)* and *offer value (cost)* are encoded in the OFC but not the ACCd (*Figure 5*). On the other hand, choice outcome in spatial (*chosen offer location*) and action-based (*chosen target location*) reference frames were represented in ACCd but not in OFC (*Figure 5*). We thus compared the strength and pattern of the variables that are commonly encoded in these two brain regions, including *chosen value*, *chosen cost,* and *chosen juice*.

We first examined the encoding of *chosen value*. We compared the percentage of *chosen value* coding neurons across the time windows of a trial. In the post-offer window, the percentage of neurons encoding the *chosen value* was strikingly similar with 13.7% and 14.3% in ACCd and OFC, respectively (*Figure 7A*). However, ACCd demonstrated more sustained encoding of *chosen value* with significantly higher percentage of neurons in late-delay (ACCd: 20.1%, OFC: 10.5 %; p < 0.005, $\chi^2$-test), mem-delay (ACCd: 13.3%, OFC: 6.3 %; p < 0.01, $\chi^2$-test), and pre-target (ACCd: 11.2%, OFC: 3.2 %; p < 0.001, $\chi^2$-test) windows (*Figure 7A*).

We next compared the percentage of *chosen cost* and *chosen juice* coding neurons between the two areas. Encoding of the variable *chosen cost* was stronger in the OFC in post-juice window (*Figure 7B*, OFC: 10.9%, ACCd: 1.9 %; p < 10$^{-4}$, $\chi^2$-test). Moreover, encoding of the variable *chosen juice* was substantially higher in OFC in all time windows following target presentation including post-target (*Figure 7C*, ACCd: 1.6%, OFC: 9.0 %; p < 10$^{-3}$, $\chi^2$-test), pre-go (OFC: 7.5%, ACCd: 0.31 %; p < 10$^{-4}$, $\chi^2$-test), pre-juice (OFC: 21.8%, ACCd: 2.2 %; p < 10$^{-10}$, $\chi^2$-test), and post-juice (OFC: 19.6%, ACCd: 3.4 %; p < 10$^{-7}$, $\chi^2$-test) windows. However, within each brain region, encoding of the binary choice outcome signal differs. In OFC, it was skewed toward the juice-based reference frame (variable *chosen juice*); in ACCd, it was heavily skewed toward the cost-based reference frame (variable *chosen cost*).

In summary, our analyses revealed that in the cost-involved decision making, ACCd is primarily engaged in encoding the value and cost associated with the chosen option.

## Discussion

In our experiments, monkeys performed economic choices between goods defined by three attributes – juice type (flavor), quantity, and action cost. Importantly, all three attributes were communicated to the animal early in the trial during offer presentation, well before the location of the eventual saccade targets were indicated. Hence animals could make their decision independently of action planning. Choices were influenced by all three attributes. In a previous study using the same choice task, the analysis of neuronal responses in central OFC (area 13 m) provided evidence that decisions were resolved in goods space. In the current study, we investigated the contribution of ACCd to this decision process. Neuronal coding in ACCd was consistent with good-based decisions. Furthermore, neurons in ACCd did not encode offer values and did not appear to encode any variable that would provide input to the ongoing decision. Conversely, neurons in ACCd encoded variables related to the choice outcome in multiple reference frames, including variables *chosen value*, *chosen offer location only*, *chosen target location only*, *chosen cost,* and *chosen juice*. Consistent with other reports on the prefrontal cortex (*Barak et al., 2013*; *Fusi et al., 2016*; *Mashhoori et al., 2018*; *Rigotti et al., 2013*), multiplexing of information on chosen cost and chosen target location was observed in time windows after target presentation. Importantly, *chosen offer location only* and *chosen target location only* were

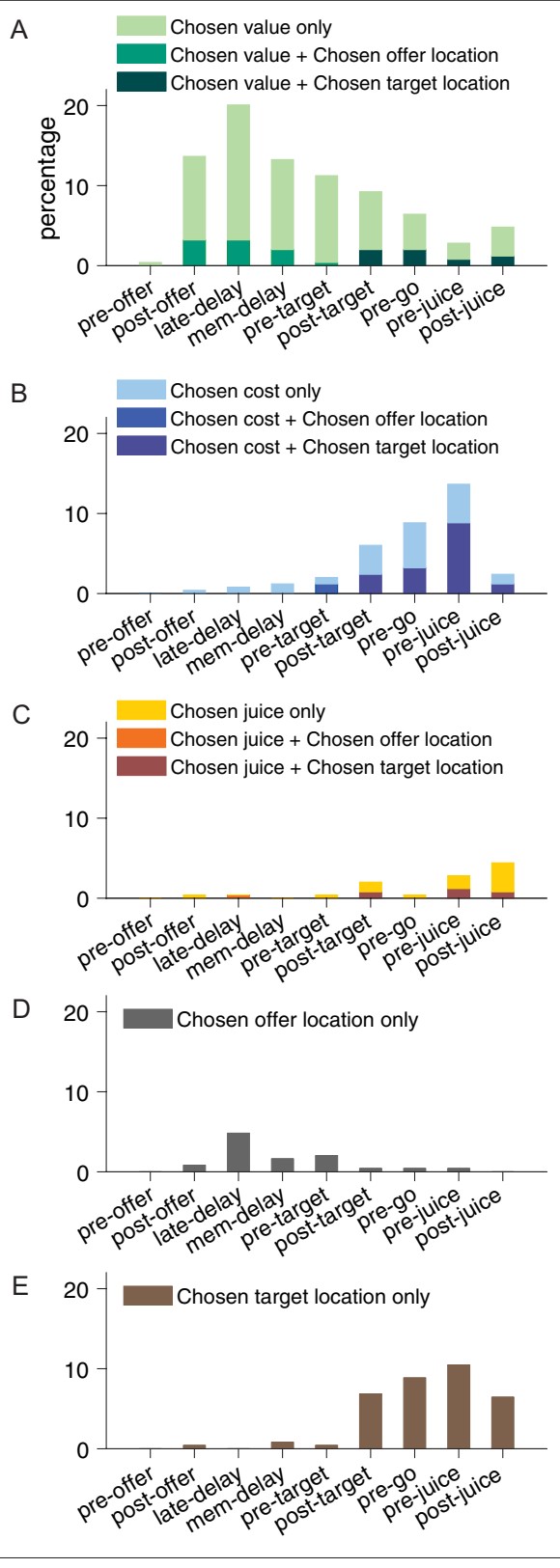

**Figure 5.** Percentage of encoded variables across task time windows. (**A**) Variable *chosen value*. (**B**) Variable *chosen cost*. (**C**) Variable *chosen juice*. Light shade indicates the percentage of neurons encoding the covariate but not the categorical variable (*chosen offer location* in time windows before target presentation or *chosen target location* in time windows after target presentation, analysis of covariance [ANCOVA], parallel model). Dark shade

*Figure 5 continued on next page*

*Figure 5 continued*
indicates the percentage of neurons encoding both the covariate and the categorical variable. (**D**) Variable *chosen offer location only*. (**E**) Variable *chosen target location only*. The same set of variables were selected by stepwise and best-subset variable selection analysis.

not significantly represented in the OFC. Thus, ACCd may function as a hub for broadcasting information associated with the choice outcome given its simultaneous representation of decision output in juice-based, cost-based, spatial-based, and action-based reference frames.

In our earlier study of anterior cingulate cortex (*Cai and Padoa-Schioppa, 2012*), we tested the role of this region in economic choice by recording neuronal activity from both the dorsal and ventral bank of the cingulate sulcus while monkeys performed a choice task involving the tradeoff of juice type and quantity. We observed that neurons in both regions encode only post-decision variables, namely *chosen value* and *chosen juice*, but not pre-decision variables in either juice-based or spatial reference frames. The results indicated that ACCd does not contribute to economic choice in the context of goods consisting of the attribute of juice taste and quantity. However, the outcome does not exclude a possible contribution of ACCd in other choice contexts, especially when choices are affected by action costs. One prevailing hypothesis is that when action costs are integrated into the computing of subjective value for choice, the neural locus of value computation and choice may include ACCd (*Hosokawa et al., 2013*). Our present results do not support this proposal. Indeed, activity in ACCd did reflect the aspect of physical effort, but not in such a form that would contribute to the encoding of subjective value.

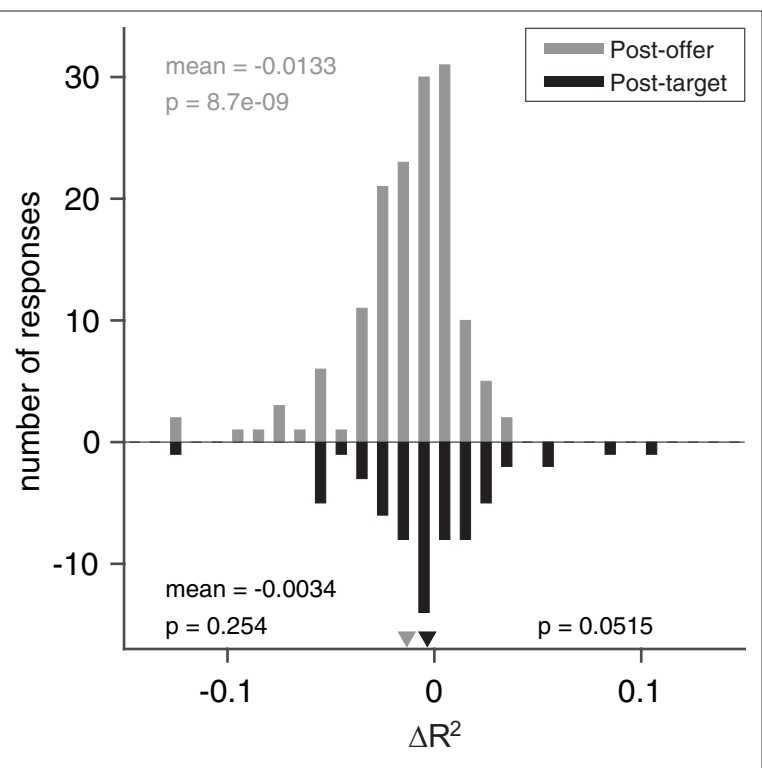

**Figure 6.** Model comparisons for chosen value responses. The x-axis represents the difference $\Delta R^2$. In this case, we examined separately early time windows (post-offer to pre-target windows, top, N = 148 responses) and late time windows (post-target to post-juice windows, bottom, N = 65 responses). In the early time windows, mean($\Delta R^2$) = –0.013 (p < 10[-8], Wilcoxon signed-rank test). In the late time windows, mean($\Delta R^2$) = –0.0034 (p = 0.25, Wilcoxon signed-rank test). The difference between the two measures was marginally significant (p = 0.052, Wilcoxon rank-sum test).

The online version of this article includes the following source data for figure 6:

**Source data 1.** Source data for *Figure 6*.

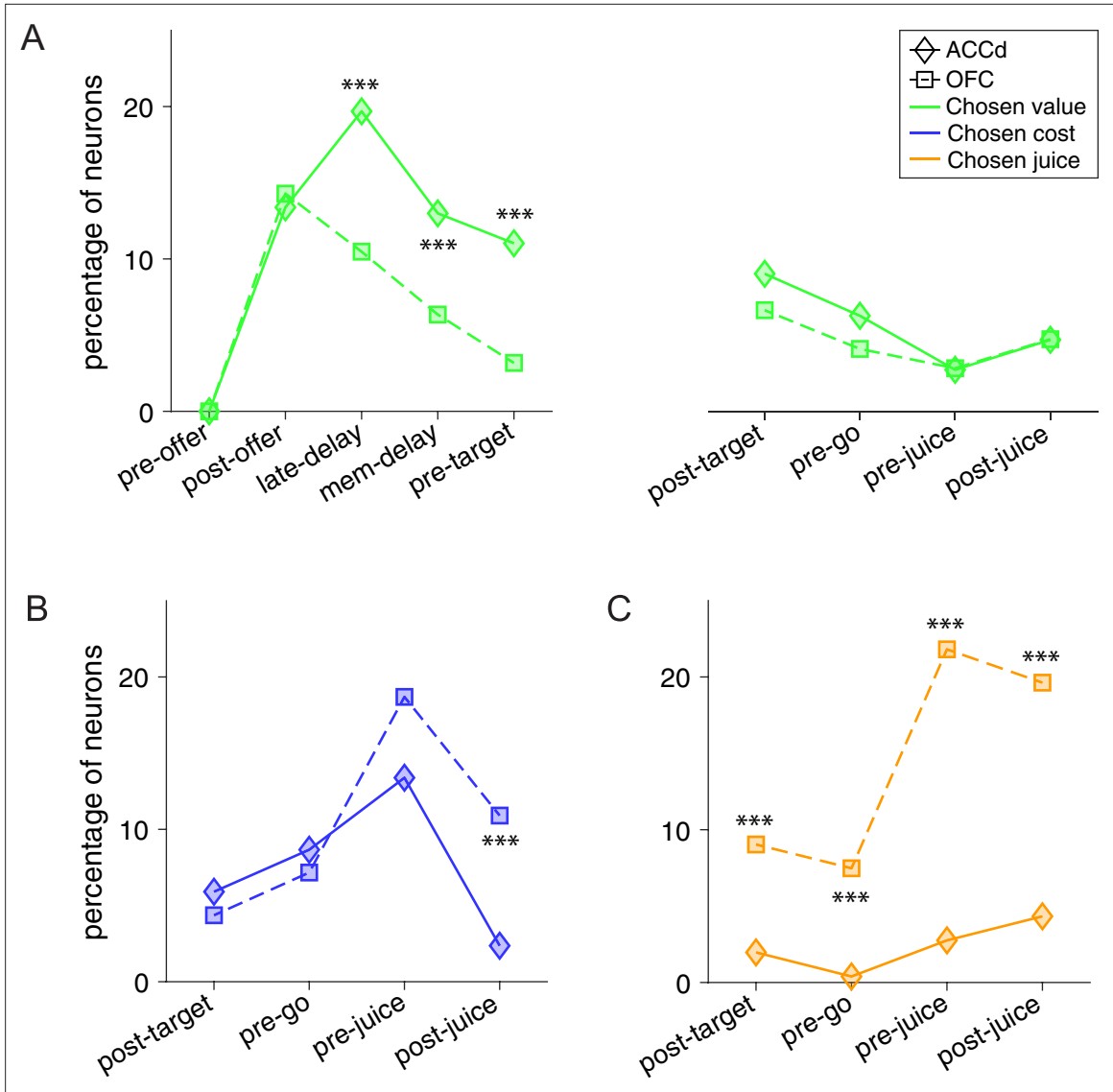

**Figure 7.** Contrasting the encoding of chosen cost and chosen juice in the dorsal anterior cingulate cortex (ACCd) and orbitofrontal cortex (OFC). (**A**) Comparison of the percentage of neurons encoding chosen value in ACCd and OFC (***, p < 0.001, $\chi^2$-test). (**B**) Comparison of the percentage of neurons encoding chosen cost in ACCd and OFC (***, p < 0.001, $\chi^2$-test). (**C**) Comparison of the percentage of neurons encoding chosen juice in ACCd and OFC (***, p < 0.001, $\chi^2$-test).

Our results provide additional evidence disambiguating the role of ACCd in effort-based decisions. One proposal argues that in effort-based decisions, ACCd integrates multiple attributes and compute the utility of an offer to guide optimal choice. This view has gained support from both single-unit studies in non-human primates and human fMRI studies (*Croxson et al., 2009*; *Hosokawa et al., 2013*; *Kennerley et al., 2009*; *Prévost et al., 2010*). However, a non-human primate study showed that lesion of ACCd did not affect the animal's choices, which require the integration of reward and effort (*Kennerley et al., 2006*). Moreover, rodent electrophysiological studies demonstrated that encoding of effort was immediately preceding the execution of effort rather than at the time of decision (*Cowen et al., 2012*; *Hashemnia et al., 2020*). Consistent with this outcome, rats with ACCd lesions were far more likely than controls to abort a high reward/high effort choice at the point where they encountered the ramp (*Holec et al., 2014*). These results suggest that the ACCd is needed when deciding to maintain a previously chosen course of action against a high effort. In this view, the ACCd encodes the amount of effort necessary to achieve a particular goal thus provides a signal that allows other systems to prepare for exerting the appropriate amount of effort. In our study, we did not find a systematic representation of the action

cost associated with any specific offer. However, there was significant encoding of the chosen action cost in the late phase of the trial, when actions were planned. Moreover, *chosen value* signals were much more sustained in ACCd. Taken together, our findings support the view of ACCd in regulating downstream systems based on the outcome of decisions.

The fact that we used the same experimental design and the same animals in our OFC and ACCd studies allowed us to compare the contribution of OFC and ACCd to economic choice under variable action costs. We first discovered that the percentage of task-relevant cells was strikingly similar in these two regions. At the same time, these two regions did not encode the same set of task-relevant variables. Most importantly, OFC but not ACCd encoded offer values in cost-based reference frame, integrating all choice-relevant attributes. In our paradigm, goods and values might be represented in the frames of reference defined by the juice type, the action cost, the spatial location of the offer, or the spatial location of the saccade target. Neurons in ACCd did not appear to encode offer values in any of these reference frames. On the other hand, neurons in ACCd encoded the saccadic action of the animal – a signal that was absent in OFC. Furthermore, encoding of the chosen value was more sustained in the ACCd as indicated by significantly higher percentage of neurons encoding this variable in late-delay and mem-delay time windows. Lastly, in ACCd, percentage of neurons encoding the *chosen cost* was substantially higher than that encoding the *chosen juice*. Our discovery that few neurons in ACCd encode *chosen juice* likely reflects the fact that it is the OFC but not the ACCd that receives direct input from the primary taste cortex (*Rolls, 2015*). Overall, these findings differentiate the ACCd from the OFC in informing the motor system of the expected reward and cost associated with the outcome of the choice.

In another study, *Hosokawa et al., 2013*, compared the neuronal coding in ACCd and OFC in a choice task involving cost-benefit tradeoff. Our findings differ in two aspects. First, *Hosokawa et al., 2013*, reported contralateral action value coding in ACCd while we did not discover significant offer value coding in either spatial- or action-based reference frames in our ACCd recordings. Second, they reported that there was no action-based value representation in the OFC, therefore concluded that OFC does not integrate action cost in economic choice. Two elements may help explain the discrepancies between our findings in ACCd and OFC (*Cai and Padoa-Schioppa, 2019*) and those of *Hosokawa et al., 2013*. First, we recall that *Hosokawa et al., 2013*, only tested value-related variables such as the benefit, cost, and discounted value in action-based reference frame. Most importantly, they did not test the variable that is related to the saccade direction, which is highly correlated with the spatial value signal. As a consequence, contralateral value signal may not be significant if chosen target location was included in their regression analysis. Indeed, in our analysis, saccade direction (or *chosen target location*) was identified as one of the variables that explained a significant portion of neuronal activity in ACCd (*Cai and Padoa-Schioppa, 2012*; *Cai and Padoa-Schioppa, 2019*). The second and often overlooked aspect is that value may be encoded in schemes other than the action-based reference frame. In their study, each unique combination of reward quantity and cost was presented by a unique picture. Thus, information on good attributes were conveyed to the animal with an 'integrated' visual representation. Accordingly, a distinct group of neurons may have been recruited to encode the reward and cost conjunctively represented by a unique fractal, which would result in 16 groups of *offer value* coding neurons.

To conclude, our findings demonstrate that choice outcome coding dominates the neuronal representations in ACCd during economic choice. When action cost was introduced as an additional value attribute, neurons in ACCd expanded the scope of encoding incorporating choice outcome signals in the cost-based reference frame. However, the detailed neuronal reorganization during this process remains unknown. To uncover this process, future studies should monitor the same population of neurons during such reorganization.

## Materials and methods
### Choice task
All experimental procedures conformed to the NIH Guide for the Care and Use of Laboratory Animals and were approved by the Institutional Animal Care and Use Committee (IACUC) at Washington University.

The experimental procedures were as described in a previous study (*Cai and Padoa-Schioppa, 2019*). Briefly, two rhesus monkeys (B, male, 9.0 kg; L, female, 6.5 kg) participated in the experiments.

Before training, a head-restraining device and an oval recording chamber were implanted on the skull under general anesthesia, as previously described (*Cai and Padoa-Schioppa, 2014*). During the experiments, animals sat in an electrically insulated enclosure (Crist Instruments) with their head restrained. The eye position was monitored with an infrared video camera (Eyelink; SR Research). The behavioral task was controlled through a custom-written software (http://www.monkeylogic.net/) based on Matlab (MathWorks).

*Figure 1* illustrates the choice task. In each session, an animal chose between two juices offered in variable amounts and at different action cost. The offers were represented by sets of color symbols, with the color indicating the juice type, the number of symbols indicating juice amount, and the shape of the symbols indicating the action cost (cross for low cost; diamond for high cost). The trial started with the animal fixating the center fixation point. After 1.5 s, two offers appeared on the two sides of the fixation point. The offers remained on the monitor for 1 s. The monkey continued fixating for another 1 s, after which two saccade targets appeared. The two saccade targets, represented by two dots with matching color to that of the two offers, were located on two concentric rings centered on the fixation point. The radius for low-cost targets was 3.5–4°; the radius for high-cost targets was 10–16°. In each trial, one of the saccade targets was placed on the low-cost (small radius) ring while the other saccade target was placed on high-cost (large radius) ring. The two targets were always placed on opposite sides of the center fixation point. The angle defining their position was selected on every trial among four possible values, corresponding to 22.5°, 157.5°, 202.5°, and 337.5° from azimuth. Thus, for each juice, there were eight possible saccade target positions (2 distances × 4 angles). The monkey maintained center fixation for a randomly variable delay (0.6–1.2 s), at the end of which the fixation point was extinguished (go signal). At that point, the animal was allowed to indicate its choice with a saccade. The animal had to maintain peripheral fixation for an additional 0.75 s, at the end of which the chosen juice was delivered. In each session, the two juice quantities varied pseudo-randomly from trial to trial. The spatial positions of the offers, the action costs, and the angle of the saccade targets varied pseudo-randomly and were counter-balanced across trials. Different pairs of juices were used across sessions.

## Neuronal recordings

Procedures for surgery, neuronal recordings, and spike sorting were similar to those described previously (*Cai and Padoa-Schioppa, 2014*). In brief, the recording chamber (main axes, 50 mm × 30 mm) was centered on stereotaxic coordinates (A30, L0), with the longer axis parallel to the coronal plane. Neuronal recordings were guided by structural MRI obtained for each animal before and after the implant. In monkey B, we recorded from both hemispheres and recording locations ranged A28-A38 in the anterior-posterior direction (with the corpus callosum extending anteriorly to A36). In monkey L, we recorded from the left hemisphere and recording locations ranged A25-A34 in the anterior-posterior direction (with the corpus callosum extending anteriorly to A31). Tungsten electrodes (125 μm diameter, FHC) were advanced using custom-built motorized micro-drives, with a 2.5 μm resolution. We typically used four electrodes in each session. Electrical signals were amplified and band-passed filtered (high pass: 300 Hz, low pass: 6 kHz; Lynx 8, Neuralynx, Inc). Action potentials were detected on-line and waveforms were saved to disk (25 kHz sampling rate; Power 1401, Spike 2; Cambridge Electronic Design). Spike sorting was conducted off-line (Spike 2; Cambridge Electronic Design) and only cells that appeared well isolated and stable throughout the session were included in the analysis.

## Analysis of choice patterns

All analyses were conducted in Matlab (MathWorks). On any given trial, one 'offer' was defined by a juice type, its quantity, and its action cost (e.g., 3B−). An 'offer type' was defined by two offers (e.g., [1A+:3B−]). In this notation, '−' indicates high action cost (long saccade) and '+' indicates low action cost (short saccade). A 'trial type' was defined by two offers and a choice (e.g., [1A+:3B−, A]). Of note, the position of each saccade target was defined by a distance (two possible values) and an angle (four possible values). For the purpose of all the analyses, we turned the angle into a binary variable corresponding to whether the target associated to juice A or the chosen target was placed in the contralateral or ipsilateral hemifield.

In the behavioral analysis, we examined several factors that could affect choices, including the juice quantity, the action cost, the outcome of the previous trial (choice hysteresis), a term capturing a visual

side bias, and a term capturing a saccade side bias. We thus constructed the following logistic model:

$$\text{choice } B = 1/(1+e^{-x})$$

$$X = a_0 q_B - a_1 q_A + a_2(\delta_{\text{juice B},+} - \delta_{\text{juice A},+}) + a_3(\delta_{n-1,B} - \delta_{n-1,A}) +$$
$$+ a_4(\delta_{\text{cost of B, cost } n-1} - \delta_{\text{cost of A, cost } n-1}) + a_5(\delta_{\text{offer B, left}} - \delta_{\text{offer A, left}}) +$$
$$+ a_6(\delta_{\text{target B, left}} - \delta_{\text{target A, left}})$$

(1)

where choice B = 1 if the animal chose juice B and 0 otherwise; $q_J$ was the quantity of juice J offered (with J = A, B); $\delta_{\text{juice J}, +}$ = 1 if juice J was offered at low cost and 0 otherwise; $\delta_{n-1, J}$ = 1 if in the previous trial the animal had chosen and received juice J and 0 otherwise; $\delta_{\text{cost of J, cost } n-1}$ = 1 if the cost of J is the same as that chosen in the previous trial and 0 otherwise; $\delta_{\text{offer J, left}}$ = 1 if the offer of juice J was placed to the left of the center fixation and 0 otherwise; and $\delta_{\text{target J, left}}$ = 1 if the saccade target associated with juice J was placed in the left hemifield and 0 otherwise. For each session, the logistic regression provided a measure for the relative value of the two juices ( $\rho$ = $a_1/a_0$), for the difference in action cost ( $\xi$ = $a_2/a_0$), for the choice hysteresis related to the chosen juice ( $\eta$ = $a_3/a_0$) and to the chosen cost ( $\phi$ = $a_4/a_0$), and for the spatial biases related to the offer position ($\delta$ = $a_5/a_0$) and to the target position ($\varepsilon$ = $a_6/a_0$). In this formulation, each factor (action cost, hysteresis, spatial biases) is quantified as a value term, and all values are expressed in units of juice B. The relative value ( $\rho$ ) is essentially the quantity ratio $q_B/q_A$ that makes the animal indifferent between the two juices. The factor $a_1$ can be thought of as an inverse temperature capturing the steepness of the sigmoid once all the effects included in the logistic regression are accounted for.

## Task-related responses

Each cell was analyzed in relation to the choice pattern recorded in the same session. In each trial, the neuronal activity was analyzed in nine time windows aligned with different behavioral events: pre-offer (0.5 s before the offer), post-offer (0.5 s after offer on), late-delay (0.5–1.0 s after offer on), mem-delay (0–0.5 s after offer off), pre-target (0.5 s before target on), post-target (0.5 s after target on), pre-go (0.5 s before the 'go'), pre-juice (0.5 s before juice delivery), and post-juice (0.5 s after juice delivery). To identify task-related responses, each neuronal response was submitted to two 3-way ANOVAs (factors [trial type × offer A location × target A location]; factors [trial type × chosen offer location × chosen target location]). We imposed a significance threshold p < 0.001. Responses that passed this criterion for at least one factor in either ANOVA were identified as 'task related' and included in subsequent analyses.

## Variable selection analysis

Our goal was to identify the variables encoded in ACCd during economic choices under variable action cost. Our strategy was to define a large number of variables neurons in this area might conceivably encode, and to use procedures for variable selection to identify a small subset of variables that would best explain the population. Our analysis combined the approach previously adopted to analyze ACCd activity during standard choices (*Cai and Padoa-Schioppa, 2012*) with that adopted for OFC activity recorded using the present task (*Cai and Padoa-Schioppa, 2019*). The variables defined *Table 2* and included in the analysis are essentially the same defined for OFC activity in this task (*Cai and Padoa-Schioppa, 2019*). Concurrently, earlier work found that many neurons in ACCd encoded the movement direction, alone or convolved with other decision variables (*Cai and Padoa-Schioppa, 2012*). Importantly, the present task dissociated the spatial location of the chosen offer from the movement direction. Thus, neuronal responses were analyzed with an ANCOVA using the variables defined in *Table 2* as regressors and grouping data by whether the chosen offer or chosen target was on the ipsilateral or on the contralateral hemifield (binary variable). For early time windows (before target presentation), the categorical variable included in the ANCOVA was *chosen offer location*; for late time windows (after target presentation), the categorical variable was *chosen target location*.

In preliminary analyses, we tested different models of the ANCOVA and observed that the interaction term was rarely significant. We thus report here only the results obtained using the 'parallel lines'

model. This model assumes that the encoding of the factor (variable) and the group (chosen location) are statistically independent. A variable was said to 'explain' the response if the factor in the ANCOVA was significant ($p < 0.05$). For each variable, the ANCOVA also provided an $R^2$. This was the total $R^2$, computed including both the factor and the group. For variables that did not explain the response, we arbitrarily set $R^2 = 0$. These criteria were used for all variables except the *chosen offer location only* and *chosen target location only*, for which we proceeded as follows. We computed the 'horizontal lines' model of the ANCOVA, which assumes that firing rates only depend on the group (chosen offer location or chosen target location). Variables *chosen offer location only* or *chosen target location only* were thus said to explain the response if the group was significant in this analysis ($p < 0.05$). From this model, we also obtained the $R^2$, which was set equal to 0 if the group was not significant. Importantly, the $R^2$ thus obtained for the variable *chosen offer location only* or *chosen target location only* was always lower than that obtained for any other variable that explained the response. Hence, for any given response, *chosen offer location only* or *chosen target location only* provided the best explanation (highest $R^2$) only if it provided the sole explanation.

The detailed procedures used in the variable selection analysis were described in previous reports (*Cai and Padoa-Schioppa, 2012*; *Cai and Padoa-Schioppa, 2019*) and we performed the analysis based on the neuronal activity across all nine time windows. Two procedures – stepwise and best-subset – identified a small number of variables that best explained the neuronal data set. The 'explanatory power' of any subset of variables was defined as the total number of responses collectively explained by those variables. In the stepwise procedure, we selected at each step the variable that provided the highest number of best fits within any time window. We then removed from the data set all the responses explained by this variable and we repeated the procedure on the residual data. The procedure was repeated until when the marginal explanatory power of any additional variable fell <2 %. In the best-subset procedure, we identified the subset of n variables that collectively provided the highest explanatory power. Importantly, the best-subset procedure warrants optimality and the two procedures applied to our data set provided identical results.

## Dimensional integration in chosen value signals

We examined the integration of multiple determinants into single value signals for *chosen value* (*juice*) responses. To do so, we defined two variants of the variable *chosen value* (*juice*) – one cost-affected and one cost-independent. We sought to assess which variant better fit neuronal responses. For each response, we considered the two $R^2$ and we computed the difference $\Delta R^2 = R^2_{cost-affected} - R^2_{cost-independent}$ and we examined the distribution for $\Delta R^2$ across the population. We did not want to bias the results in favor of either variant. Thus, for this analysis we identified neuronal responses encoding the *chosen value* (*juice*) as follows. For each response and each value variable, we considered the two $R^2$ obtained from the two variants, and we assigned the maximum $R^2$ to the response. We then assigned each response to one of the selected variables accordingly.

## Acknowledgements

We thank Heide Schoknecht for help with animal training.

## Additional information

### Funding

| Funder | Grant reference number | Author |
| --- | --- | --- |
| National Institute of Mental Health | R01-DA032758 | Camillo Padoa-Schioppa |
| National Natural Science Foundation of China | grants 31571102 and 91632106 | Xinying Cai |
| Ministry of Education Program of Introducing Talents of Discipline to Universities | Base B16018 | Xinying Cai |

| Funder | Grant reference number | Author |
|---|---|---|
| NYU-ECNU Institute of Brain and Cognitive Science at NYU Shanghai | Joint Research Institute Seed Grants for Research Collaboration | Xinying Cai |
| Science and Technology Commission of Shanghai Municipality | grants 15JC1400104 and 16JC1400101 | Xinying Cai |
| The Shanghai Municipal Science and Technology Major Project | 2018SHZDZX05 | Xinying Cai |

The funders had no role in study design, data collection and interpretation, or the decision to submit the work for publication.

### Author contributions

Xinying Cai, Conceptualization, Data curation, Formal analysis, Investigation, Methodology, Writing – original draft, Writing – review and editing; Camillo Padoa-Schioppa, Conceptualization, Funding acquisition, Methodology, Project administration, Resources, Supervision, Writing – review and editing

### Author ORCIDs
Xinying Cai (ID) http://orcid.org/0000-0003-4997-9793
Camillo Padoa-Schioppa (ID) http://orcid.org/0000-0002-7519-8790

### Ethics

All experimental procedures conformed to the NIH Guide for the Care and Use of Laboratory Animals and were approved by the Institutional Animal Care and Use Committee (IACUC) at Washington University.

### Decision letter and Author response
Decision letter https://doi.org/10.7554/eLife.71695.sa1
Author response https://doi.org/10.7554/eLife.71695.sa2

## Additional files

### Supplementary files
• Transparent reporting form

### Data availability
All data generated or analysed during this study are included in the manuscript and supporting files.

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
