## [Decision Letter]

**Acceptance summary:**

In this study, Cai and Padoa-Schioppa examined how effort is integrated into economic decisions by recording neural activity from the dorsal anterior cingulate cortex (ACC) in monkeys making choices based on costs and benefits. The main results provide evidence against the notion that ACC contributes to evaluation of potential actions. Instead, neurons predominantly coded for post-decision variables, such as cost of the chosen target and the juice type of the chosen offer, but not pre-decision variables, such as offer values. This is in contrast to the orbitofrontal cortex, where neurons encoded effort associated with choice options. Together, the results are convincing and highlight potentially unique roles of ACC neurons in learning and decision making.

**Decision letter after peer review:**

Thank you for submitting your article "Neuronal Activity in Dorsal Anterior Cingulate Cortex during Economic Choices under Variable Action Costs" for consideration by *eLife*. Your article has been reviewed by 3 peer reviewers, and the evaluation has been overseen by Erin Rich as the Reviewing Editor and Joshua Gold as the Senior Editor. The following individual involved in review of your submission has agreed to reveal their identity: Alicia Izquierdo (Reviewer #2).

Essential revisions:

Overall, the reviewers felt that this study provides important new information about primate ACC. The behavioral task is powerful and the analyses of both the choice behavior and neural data are rigorous. The fact that the results in the present study can be directly compared to OFC broadens the interpretation and importance of the study. In discussion, there were four points that the reviewers deemed most important to address. Further detail on each can be found in the individual reviewer comments appended below. Please be sure that your revised submission addresses the following:

1) The first essential revision is an exploration of whether any of the variables used in this study correlate with decision conflict, and whether conflict is represented in the neuronal responses.

2) It was felt that a strength of the study is the contrast between ACC and OFC, and this should be highlighted more in the manuscript, including the addition of formal comparisons with figures.

3) Third, further explanation of the variable selection methods are requested.

4) Finally, it was felt that the emphasis in the discussion of Hosokawa et al., could be shortened.

*Reviewer #1 (Recommendations for the authors):*

Specific comments

The contrast between Figure 4A and B is striking, in that 4A shows encoding of pre-choice variables (offer value juice, offer value cost, etc.) that largely disappear in 4B. If I understand the variable selection approach correctly, this could be due to correlations among the predictors in the model. Could the authors clarify whether this is a reasonable interpretation and provide a figure or table showing these correlations? If this is not the case, how is the encoding of pre-choice variables commensurate with the conclusions of the study? A second possibility is that the difference between 4A and B is driven in part by reducing each neuron's encoding to only one of the predictors. Given the extensive literature on mixed selectivity, particularly in frontal neurons, if this approach was used, couldn't it remove secondary variables that are legitimately encoded by these neurons?

In Figure 6, is it possible to assess whether the same neurons shift from not encoding effort to incorporating it in the later epoch, versus whether new neurons are recruited? Also, can the authors report the statistical cut-off used to include a neuron in this analysis?

The differences between OFC and ACC encoding offer value variables are discussed qualitatively, but it would be helpful to add formal comparisons to Figure 7.

The relative lack of chosen juice coding in ACC versus OFC is interesting. Could the authors discuss possible reasons or implications of this finding?

*Reviewer #2 (Recommendations for the authors):*

In Table 1 authors show neural responses that were identified as task-related and that 25% of the neurons were modulated by "trial type." Of the 249 cells that were modulated by at least one factor (spatial configuration of offer, target, location of chosen offer, etc), to have 25% modulated by trial type is a very large proportion of modulated neural responses. Authors briefly describe trial type as "two offers and a choice" but it is not at all clear how it relates to reference frame and/or time window- this should be clarified.

Authors could cite other relevant theoretical perspectives and previous work that are much in alignment with their present findings: for example on mixed selectivity to produce higher dimensionality for complex cognition (Fusi et al. Curr Op Neurobiol 2016) and multiplexing of spatial position and choice related information in ACC (Mashhoori et al. 2018 eLife).

---

## [Author Response]

Essential revisions:Overall, the reviewers felt that this study provides important new information about primate ACC. The behavioral task is powerful and the analyses of both the choice behavior and neural data are rigorous. The fact that the results in the present study can be directly compared to OFC broadens the interpretation and importance of the study. In discussion, there were four points that the reviewers deemed most important to address. Further detail on each can be found in the individual reviewer comments appended below. Please be sure that your revised submission addresses the following:1) The first essential revision is an exploration of whether any of the variables used in this study correlate with decision conflict, and whether conflict is represented in the neuronal responses.

This point is related to the following comment from Reviewer 3:

“Do any of the variables used in this study correlate with a conflict? When the authors previously studied ACC, they discarded the conflict monitoring hypothesis – a hypothesis that is well established for ACC hemodynamic responses – for ACC single cell activity based on neural data from 'difficult' decisions (Cai and Padoa- Schioppa, 2012). The definition of difficulty they used, then, was descriptive and based on reaction times (RTs). They defined the most difficult trials as those trials with the longest RTs and discovered that those trials had options with similar offer values. This definition of choice difficulty appears to be contrived from evidence accumulation models/tasks, where normatively harder judgments elicit longer RTs. However, there is no normative economic reason that trials with similar offer values are more difficult or should cause conflict. After all, according to theory, choosing between two options with the same value is as easy as flipping a coin. Here, it seems like the authors could have a more fitting definition of conflict. For example, conflict can be operationalized by considering trials when the animal must choose between a high value/high-cost option and a low-value/low-cost option. In that case, the costs and benefits are in conflict. What do the RTs look like? Do the RTs indicate conflict resolution? If so, is this reflected in neuronal responses?”

We thank the reviewer for raising this important point. First, we would like to clarify that both in this study and in our previous study of ACC (Cai and Padoa-Schioppa 2012) we imposed a delay between offer presentation and the go signal. Such delay is critical to disentangle value comparison from action selection. However, the delay effectively dissociates reaction times from the decision difficulty. Normally, we operationalize the decision difficulty (or conflict) with the variable value ratio = chosen value / unchosen value. In an early behavioral study conducted in capuchin monkeys, where no delay was imposed between offer presentation and the go signal, we found that reaction times were strongly correlated with the value ratio, as one would naturally expect (Padoa-Schioppa, Jandolo et al., 2006). In the previous study of ACC (Cai and Padoa-Schioppa 2012) we referenced that earlier result but, again, we did not analyze reaction times.

Coming to the present study, we addressed this question by including in the variable selection analyses the two variables value ratio and cost/benefit conflict = cost of A * sign (offer value A – offer value B) (see also Table 2). The results of the updated analysis are illustrated in the new Figure 4. In essence, including these two variables did not affect the results of the variable selection analysis. That is, both the stepwise and best-subset methods selected the variables chosen value, chosen cost, chosen juice, chosen offer location only and chosen target location only.

2) It was felt that a strength of the study is the contrast between ACC and OFC, and this should be highlighted more in the manuscript, including the addition of formal comparisons with figures.

This point is related to the following comment from Reviewer 1:

“The differences between OFC and ACC encoding offer value variables are discussed qualitatively, but it would be helpful to add formal comparisons to Figure 7.”

We thank Reviewer 1 for appreciating the strength of this study in contrasting between ACC and OFC. We did not include the variables offer value (juice) or offer value (cost) in the analysis of Figure 7 in the original analysis because neither variable was picked up by the variable selection analysis. However, to validate the conclusions, we have now analyzed these two variables as if they were selected by the variable selection analysis. We thus conducted two analyses to compare the percentage of neurons encoding offer value (juice) and offer value (cost) in ACCd and OFC. First, for each brain region, we performed a binomial test to assess whether the percentage of cells encoding either variable was significantly above chance level (5%). Second, for each variable – offer value (juice) and offer value (cost) – we performed a Chi-squared test to assess whether the percentage of cells encoding it differed significantly across areas. The outcomes of these analyses are summarized in the Author response table 1 and Author response table 2. For offer value (juice), the percentage of encoding neurons exceeded chance level in OFC (before target presentation) but not in ACC. Moreover, the percentage of neurons encoding offer value (juice) in OFC was significantly higher than that in ACCd in all time windows before target presentation. For offer value (cost), the encoding was significantly above chance only in OFC (post-offer window). Furthermore, the percentage of cells encoding this variable was significantly higher in OFC than in ACC (post-offer and late-delay time windows). In conclusion, the difference between the two areas was statistically significant.

**Author response table 1. sa2table1:** 

	* **Offer value (juice)** *						
	**ACCd**	**OFC**					
	Number	Percentage	p (binomial test)	Number	Percentage	p (binomial test)	p (chi-square test)
**Post-offer**	5	2.0	0.9953	60	18.9	0	0
**Late-delay**	4	1.6	0.9987	44	13.9	0	0
Mem-delay	1	0.40	1	26	8.2	0.0099	0
Pre-target	0	0	1	17	5.4	0.4188	0.0002

**Author response table 2. sa2table2:** 

	* **Offer value (cost)** *						
	**ACCd**	**OFC**					
	Number	Percentage	p (binomial test)	Number	Percentage	p (binomial test)	p (chi-square test)
**Post-offer**	3	1.2	0.9997	27	8.5	0.0054	0.0001
**Late-delay**	4	1.6	0.9987	20	6.3	0.1719	0.0059
Mem-delay	3	1.2	0.9997	8	2.5	0.9905	0.2592
**Pre-target**	6	2.4	0.9865	10	3.2	0.9573	0.5956

3) Third, further explanation of the variable selection methods are requested.

This point is related to the following two comments from Reviewer 1:

“The contrast between Figure 4A and B is striking, in that 4A shows encoding of prochoice variables (offer value juice, offer value cost, etc.) that largely disappear in 4B.

If I understand the variable selection approach correctly, this could be due to correlations among the predictors in the model. Could the authors clarify whether this is a reasonable interpretation and provide a figure or table showing these correlations?”

Indeed, the main reason why offer value variables largely disappeared in Figure 4B is that all these variables were correlated with the variable chosen value, and chosen value generally provided a better fit. Author response image 1 illustrates this point by showing the distribution of Pearson’s correlation between each offer value variable and the chosen value across sessions. (Note that the correlation depended on the set of offer types and on the animal’s choices). Notably, variables were substantially correlated in most sessions.

**Author response image 1. sa2fig1:** 

“A second possibility is that the difference between 4A and B is driven in part byreducing each neuron's encoding to only one of the predictors. Given the extensive literature on mixed selectivity, particularly in frontal neurons, if this approach was used, couldn't it remove secondary variables that are legitimately encoded by these neurons?”

To test this possibility, we took the same approach as that in Padoa-Schioppa and Assad (2006). Consider a response encoding variable X at the first order (i.e., a response explained by X better than by any other selected variable), to establish whether adding a second variable Y to the regression provides a significantly better account, we compute F_Y|X_ = (n-3) * (R^2^_XY_ – R^2^_X_) / (1- R^2^_XY_). In the equation, R^2^_X_ is obtained from the linear regression on X only; R^2^_XY_ is obtained from the bi-linear regression on X and Y; and n is data points in the regression. We compute for each of the variables Y we wish to test as potentially encoded as the second variable, and we take the maximum F = max{F_Y|X_}. We then set a threshold F corresponding to a desired p = 0.05/N (N = number of potential second order variables). If F passes the criterion, we identify the second order variable encoded by the response. If F does not pass the criterion, we conclude that the response does not encode other variables (of those we defined and analyzed). For each neuron, chosen value was set as variable X because chosen value is the dominant variable encoded by ACCd, and the rest of the variables as Y. In Author response image 2, the first column indicates the number of responses that provide the best fit for chosen value in a single linear regression (y = a_0_ + a_1_*chosen value). The other columns indicate the number of responses that encode a secondary variable in addition to chosen value. For responses encoding chosen value, there was no consistent encoding of a secondary variable.

4) Finally, it was felt that the emphasis in the discussion of Hosokawa et al., could be shortened.

We shortened the discussion from 3 paragraphs to 1 paragraph as follows.

“In another study, Hosokawa, Kennerley et al., (2013) compared the neuronal coding in ACCd and OFC in a choice task involving cost-benefit tradeoff. Our findings differ in two aspects. […] Accordingly, a distinct group of neurons may have been recruited to encode the reward and cost conjunctively represented by a unique fractal, which would result in 16 groups of offer value coding neurons.”

References

Cai, X. and C. Padoa-Schioppa (2012). "Neuronal encoding of subjective value in dorsal and ventral anterior cingulate cortex." J Neurosci 32(11): 3791-3808.

Cai, X. and C. Padoa-Schioppa (2019). "Neuronal evidence for good-based economic decisions under variable action costs." Nat Commun 10(1): 393.

Hosokawa, T., S. W. Kennerley, J. Sloan and J. D. Wallis (2013). "Single-neuron mechanisms underlying cost-benefit analysis in frontal cortex." J Neurosci 33(44): 17385-17397.

Kennerley, S. W., A. F. Dahmubed, A. H. Lara and J. D. Wallis (2009). "Neurons in the frontal lobe encode the value of multiple decision variables." J Cogn Neurosci 21(6): 1162-1178.

Kennerley, S. W. and J. D. Wallis (2009). "Evaluating choices by single neurons in the frontal lobe: outcome value encoded across multiple decision variables." Eur J

Neurosci 29(10): 2061-2073.

Padoa-Schioppa, C., L. Jandolo and E. Visalberghi (2006). "Multi-stage mental process for economic choice in capuchins." Cognition 99(1): B1-B13.

Reviewer #1 (Recommendations for the authors):Specific commentsThe contrast between Figure 4A and B is striking, in that 4A shows encoding of pre-choice variables (offer value juice, offer value cost, etc.) that largely disappear in 4B. If I understand the variable selection approach correctly, this could be due to correlations among the predictors in the model. Could the authors clarify whether this is a reasonable interpretation and provide a figure or table showing these correlations? If this is not the case, how is the encoding of pre-choice variables commensurate with the conclusions of the study? A second possibility is that the difference between 4A and B is driven in part by reducing each neuron's encoding to only one of the predictors. Given the extensive literature on mixed selectivity, particularly in frontal neurons, if this approach was used, couldn't it remove secondary variables that are legitimately encoded by these neurons?

Please refer to Essential Revisions point #3.

In Figure 6, is it possible to assess whether the same neurons shift from not encoding effort to incorporating it in the later epoch, versus whether new neurons are recruited? Also, can the authors report the statistical cut-off used to include a neuron in this analysis?

We thank the Reviewer for raising this interesting point. For population level analysis, we rephrased the question in the following way – does chosen value tend to be encoded by the same or separate population of neurons in time windows pre and post target presentation? To this end, we identified the number of neurons that encoded chosen value in at least one of post-offer windows as well as the number of neurons encoding chosen value in at least one of the post-target windows. We then constructed the following contingency table, Author response image 3, (the total number of modulated neurons is 249) and performed a Chi-square test to assess the independence of coding. We are unable to reject the hypothesis that chosen value was encoded by separate populations of neurons before and after target presentation (odds ratio = 1.41; p = 0.27; Fisher’s exact test).

**Author response image 3. sa2fig3:** 

Criteria for including a response in the analysis is described in detail in the Methodsection (line 455 to 471). For more clarity, we modified the text as the follows (line 198 – 201).

“For each response, we considered the two R2 obtained from the two variants of chosen value variable both “explaining” the response (p < 0.05), and we assigned the maximum R2 to the response. We then assigned each response to one of the selected variables accordingly.”

The differences between OFC and ACC encoding offer value variables are discussed qualitatively, but it would be helpful to add formal comparisons to Figure 7.

Please refer to Essential Revisions point #2.

The relative lack of chosen juice coding in ACC versus OFC is interesting. Could the authors discuss possible reasons or implications of this finding?

Yes, we added a brief discussion (line 314 – 316):

“Our discovery that few neurons in ACCd encode chosen juice likely reflects the fact that it is the OFC but not the ACCd that receives direct input from the primary taste cortex.”

Reviewer #2 (Recommendations for the authors):In Table 1 authors show neural responses that were identified as task-related and that 25% of the neurons were modulated by "trial type." Of the 249 cells that were modulated by at least one factor (spatial configuration of offer, target, location of chosen offer, etc), to have 25% modulated by trial type is a very large proportion of modulated neural responses. Authors briefly describe trial type as "two offers and a choice" but it is not at all clear how it relates to reference frame and/or time window- this should be clarified.

We apologize for the confusion. We made the following revisions to improve clarity.

1) We added on line 105, “Thus, a “trial type” encompasses task-related factors in goods space.”

2) In Table 1, 25% (172/688) refers to the percentage of cells that were modulated by “trial type” in at least one of the analyzed time windows. To make it clearer, we added on line 163 “out of the total 688 neurons”.

Authors could cite other relevant theoretical perspectives and previous work that are much in alignment with their present findings: for example on mixed selectivity to produce higher dimensionality for complex cognition (Fusi et al. Curr Op Neurobiol 2016) and multiplexing of spatial position and choice related information in ACC (Mashhoori et al. 2018 eLife).

We added these references (lines 260 – 263).